# Infection with a Recently Discovered Gammaherpesvirus Variant in European Badgers, *Meles meles*, is Associated with Higher Relative Viral Loads in Blood

**DOI:** 10.3390/pathogens11101154

**Published:** 2022-10-06

**Authors:** Ming-shan Tsai, Sarah François, Chris Newman, David W. Macdonald, Christina D. Buesching

**Affiliations:** 1Wildlife Conservation Research Unit, Recanati-Kaplan Centre, Department of Biology, University of Oxford, Tubney, Oxfordshire OX13 5QL, UK; 2Department of Biology, University of Oxford, Peter Medawar Building, South Park Road, Oxford OX1 3SY, UK; 3Department of Biology, Irving K. Barber Faculty of Science, The University of British Columbia, Okanagan, Kelowna, BC V1V 1V7, Canada

**Keywords:** red queen hypothesis, coevolution, sexually transmitted infection, one health, genetic epidemiology

## Abstract

Herpesviruses are ubiquitous pathogens infecting most animals. Although host immunity continually coevolves to combat virulence, viral variants with enhanced transmissibility or virulence occasionally emerge, resulting in disease burdens in host populations. Mustelid gammaherpesvirus 1 (MusGHV-1) is the only herpesvirus species identified thus far in European badgers, *Meles meles*. No MusGHV-1 associated pathomorbidity has been reported, but reactivation of MusGHV-1 in genital tracts is linked to impaired female reproductive success. An analysis of a short sequence from the highly conserved DNA polymerase (DNApol) gene previously identified two variants in a single host population. Here we compared genetic variance in blood samples from 66 known individuals of this same free-ranging badger population using a partial sequence comprising 2874 nucleotides of the DNApol gene, among which we identified 15 nucleotide differences resulting in 5 amino acid differences. Prevalence was 86% (59/66) for the common and 17% (11/66) for the novel variant, with 6% (4/66) of badgers presenting with coinfection. MusGHV-1 variants were distributed unevenly across the population, with individuals infected with the novel genotype clustered in 3 of 25 contiguous social groups. Individuals infected with the novel variant had significantly higher MusGHV-1 viral loads in their blood (*p* = 0.002) after adjusting for age (juveniles > adults, *p* < 0.001) and season (summer > spring and autumn, *p* = 0.005; mixed-effect linear regression), likely indicating higher virulence of the novel variant. Further genome-wide analyses of MusGHV-1 host resistance genes and host phenotypic variations are required to clarify the drivers and sequelae of this new MusGHV-1 variant.

## 1. Introduction

Under the concept of ‘One Health’ (a framework recognising the interconnection between human, animal and environmental health), emerging pathogen strain monitoring and pathotype identification are considered essential for disease surveillance and prevention, not only in humans and domestic animals but also in wildlife [1,2]. The majority (60%) of emerging infectious diseases are of animal origin, predominantly (72%) from wildlife [1]. Due to a lack of fidelity during replication, viruses have a higher mutation rate than cellular organisms. This results in higher evolutionary plasticity, which favours viral adaptation to changing environments (i.e., novel hosts, new host immune strategies, different medical treatments, etc.) and the emergence of new strains [3,4]. The resulting arms race between pathogen virulence and host resistance continually shapes the evolution of competitive coevolving genes or organisms through repeated cycles of adaptation and counter-adaptation, a process formalised under the Red Queen hypothesis (RQH) [5]. When applied to host–pathogen coevolution, the RQH predicts that, under natural conditions, changes in host allele frequencies within a population cause pathogens to adjust allele frequencies to match those of the host and vice versa [6,7,8]. This can drive intrahost speciation (i.e., several closely related virus strains or species can evolve independently within the same host species) [9,10]. A recent study shows that this so-called ‘duplication’ has occurred more frequently than expected in the evolution of herpesviruses (previously believed to be dominated by cospeciation with their host species), especially among viruses belonging to the *Gammaherpesvirinae* subfamily [11].

European badgers (*Meles meles*) are promiscuous, polygynandrous mustelids [12]. Their social system is highly flexible [13], ranging from solitary or pair-living [14] to the formation of extended family groups comprising upwards of 30 individuals [15] Mustelid gammaherpesvirus 1 (MusGHV-1) is the only gammaherpesvirus known to infect badgers [16]. Like other gammaherpesviruses, after primary infection, MusGHV-1 may establish latency in white blood cells [17]. Screening badger blood samples revealed a 98–100% prevalence of MusGHV-1 in wild British and Irish populations [18,19]. Reactivation from latency causes viral replication and shedding in the mucosa in oral, nasal, and/ or genital tracts. Depending on the season and population [20], the prevalence of genital shedding of MusGHV-1 ranges from 55% to 62% [21,22]. No specific signs or pathomorbidity have been linked to MusGHV-1 reactivation in badgers. However, viral shedding is associated with lower female reproductive success and lesions in the genital tract of both sexes [21]. In our previous analysis of partial DNA polymerase (DNApol) gene sequences [20], we identified two distinct variants through the genotyping of MusGHV-1 in a single closed wild badger population located in Wytham Woods, Oxfordshire (for details of the study population, sees Buesching et al., 2010 [23]).

Since the herpesviruses DNApol gene encodes exonuclease and polymerase proteins essential for virus replication and viral genome proofreading, its quantified expression could then be used as a proxy for determining virulence [24]. The DNApol gene is thus one of the most conserved genes in herpesviral genomes and likely to be under strong purifying selection with a low substitution rate. Nevertheless, any mutations found in these domains are typically associated with lower replication fidelity (see review: [25]), contributing to higher mutation rates. If some of these mutations are beneficial to virus replication efficiency, they can, in turn, result in faster evolution rates [26]. Consequently, herpesvirus strains belonging to the same species that show differences in their DNApol genes can result in different disease expressions and outcomes in their hosts. For example, a single nucleotide difference in the equine herpesvirus 1 (EHV-1) polymerase gene on amino acid position 752 (N to D) is associated with neurological disease outbreaks [27,28,29] and higher viraemia levels in infected horses compared with those infected with the EHV-1 N_752_ strain, suggesting a higher virus replication efficiency [30]. Differences in host susceptibility to the two variants have not yet been explored, but simultaneous infections with both strains have been reported [31,32].

Here we investigated genetic, epidemiological, and phenotypic differences between the two previously reported MusGHV-1 variants in a natural host population [20]. Specifically, we aimed to:(1)acquire longer sequences of the MusGHV-1 DNA polymerase gene to investigate for additional mutations/genetic differences between the two variants;(2)establish the relative prevalence of the two MusGHV-1 variants within the population, investigate the rate of coinfection, and whether potential differences in variant prevalence are associated with demographic (i.e., host sex and age) and/or socio-geographic (i.e., badger sett and social group) patterns;(3)analyse potential differences in viral loads in blood samples between individuals infected with each or both MusGHV-1 variants.

## 2. Materials and Methods

### 2.1. Animal Sampling

We captured free-ranging badgers in Wytham Woods, Oxfordshire, United Kingdom, (51°46′26″ N, 1°19′19″ W; for details of the study site, see [33]; for details of badger trapping and handling protocols see [20,34]) in spring (21st May to 2nd June), summer (3rd to 15th September) and autumn (13th to 24th November) 2018. In summary, individual badgers were identified by their serial tattoo numbers, given at first capture as part of a long-term badger research programme [15,35]. After sedation with ketamine hydrochloride [36], oral, nasal, genital, and rectal swab samples were collected using wooden shafted cotton tips (Dynarex). Blood samples (not exceeding 2 mL) were collected by jugular venipuncture in heparin-coated tubes and stored immediately at −20 °C. All protocols and procedures were approved by the Zoology Department’s (University of Oxford) Animal Welfare and Ethical Review Body and were conducted under the Animals (Scientific Procedures) Act, 1986 (PPL: 30/3379).

### 2.2. Genotyping of MusGHV-1 DNA

DNA was extracted and purified from swab samples using DNeasy Blood and Tissue Kits (Qiagen), and from 200 μL whole blood samples using Monarch Genomic DNA Purification Kits (New England Biolabs Inc., Ipswich, MA, USA) following manufacturers’ instructions (for full methodological details see Tsai et al. [20]). MusGHV-1 DNA was amplified from each sample by PCR using primers (pol3-F and pol3-R, Table 1), specifically targeting the MusGHV-1 DNA polymerase gene. After checking amplification results on 1.5% agarose gels following electrophoresis, we sent 66 amplified PCR products for genotyping using Sanger sequencing (Zoology Sequencing, University of Oxford, Oxford, UK). Variant allocation was determined using ClustalW alignment (with the respective default parameters) in MEGA11 (Molecular Evolutionary Genetics Analysis version 11.0.8, Tamura, Stecher, and Kumar) according to nine substitutions across 691 nucleotides identified between the two previously reported MusGHV-1 sequences (accession numbers MT332100 and MT332101) in the same study population: at position 363 (C to T), 378 (T to C), 383 (A to G), 480 (T to C), 486 (T to C), 537 (A to G), 540 (G to A), 546 (G to A), and 558 (C to A) [20]. Coinfection was reported if the aforementioned signals for both nucleotides were present in a given sequencing product.

### 2.3. MusGHV-1 DNA Polymerase Gene Mapping and Phylogenic Analysis

One sample of each variant was chosen for extended DNApol gene genotyping using primer pairs designed specifically for this study (Table 1). The same PCR procedures and conditions were used as described above, and successfully amplified PCR products were sent for Sanger sequencing (Zoology Sequencing, University of Oxford). Sequences were then mapped to the reference gene (DNA polymerase gene of MusGHV-1, Accession number: AF275657), assembled, translated to amino acids, and aligned using the ClustalW method (with set default parameters) [37] to 7 representatives of published gammaherpesvirus sequences in GenBank (having either complete or > 1000 bp of the DNA polymerase gene), using the software Unipro UGENE (version 37, Okonechnikov, Golosova, Fursov, the UGENE team, Novosibirsk, Russia) [38]. For each isolate, a purifying selection test using the Nei-Gojobori method was employed to determine whether the number of nonsynonymous substitutions per nonsynonymous site (dNS) was fewer than the number of synonymous substitutions per synonymous site (dS) [39]. Nucleotide sequences were aligned using ClustalW with default parameters. A maximum likelihood (ML) phylogenetic tree was constructed under the GTR + I + gamma nucleotide model using 100 bootstrap replicates using MAFFT version 7.450 [40]. The ML tree was mid-point rooted.

### 2.4. Geographic Distribution of MusGHV-1 Strains and Social Group-Specific Prevalence

The social group affiliation of each individual was determined using previously established residency rules [41,42] (Appendix A), with social group ranges established through bait-marking surveys [34,42]. Social group-specific MusGHV-1 variant prevalence was calculated by dividing the number of total positive badgers by the total of tested badgers resident in each social group. All maps were drawn using QGIS (version 3.14.15, QGIS Development Team).

### 2.5. Blood Viral Load Quantification

We selected blood samples from individuals with known MusGHV-1 variant infections. Some individuals (see Table 2) were sampled in more than one trapping season. We analysed each sample’s relative MusGHV-1 genome copy numbers using StepOnePlus PCR Systems (Applied Biosystems, ABI) to obtain Ct values. We used primers designed by Sin et al. [18] targeting the MusGHV-1 DNA polymerase gene (forward: 5′-GGAGAGTGCTGACCGATGGA; reverse: 5′-AAAAGCCTGGAATTGGATCAATAA, product length: 150 bp). We prepared 20 μL of reaction mix containing 0.5 mM of each primer, 10 μL of Luna Universal qPCR Master Mix, 3 μL of RNase-free water and 5 μL of DNA template. Amplification conditions were as follows: 95 °C for 60 s (initial denaturation), 45 cycles of 95 °C (denaturation) for 15 s, and 60 °C for 30 s (extension). StepOne Software (version 2.3) obtained Ct values of each sample automatically.

### 2.6. Statistical Analyses

All statistical analyses were performed using R studio (version 1.2.1335, RStudio Team, Boston, MA, USA) [44,45]. Fisher’s exact tests were used to test if infections with the MusGHV-1 variant differed with sex, age group (adult: ≥2 years old, and juvenile: <2 years old) or social group (*n* = 25). Differences in viral loads between variants were analysed with a mixed-effect linear regression analysis using the R package lmerTest. We used Ct values as the response variable and variants (common, coinfection and novel), age group and sampling seasons (spring, summer and autumn) as fixed effects. Tattoo ID was included as a random effect, as some individuals were sampled more than once.

## 3. Results

### 3.1. Substitutions in the DNApol Gene of the Two MusGHV-1 Variants

We sequenced 66 MusGHV-1 PCR products amplified from the partial DNApol gene. All sequences were trimmed to 693 base pairs and were confirmed to be MusGHV-1, showing between 98.7% and 100% identity with the published MusGHV-1 sequence isolated from a badger in Cornwall, England (Accession number: AF275657).

The extended genotyping of the DNApol gene resulted in the acquisition of a total sequence length of 2874 nucleotides for each of the two MusGHV-1 variants. This enabled us to test for additional variations between the two variants. In total, we found 15 nucleotides and five amino acid differences in the DNA polymerase domain between the two variants, which we classified here as the ‘common variant’ and the ‘novel variant’, as the former was almost identical (99.9% nucleotide identity) to the reference sequence. In contrast, the novel variant only shared 99.4% identity with the reference sequence. Both MusGHV-1 variants had three nucleotides and two amino acid substitutions in common with the reference sequence, all located at the exonuclease domain (details of these substitutions are listed in Table 3 and Figure 1). There was support for strong purifying selection between the two variants in the Wytham badger population (dNS < dS, *p* value = 0.004, dS/dNS = 2.66).

Among these 66 individuals, the common variant was detected in 59 (89.4%), including genital swabs (n = 49), oral swabs (*n* = 5), rectal swabs (*n* = 2), and blood samples (*n* = 5). The novel variant was detected in 11 (16.7%) individuals, including in genital swabs (*n* = 9) and blood samples (*n* = 5). Coinfection was confirmed in four individuals (6.1%) in three blood samples and one rectal swab (Appendix A).

### 3.2. Phylogenetic Relationships between the Two MusGHV-1 Variants and Other Gammaherpesvirus Species

All Carnivora-infecting gammaherpesviruses are grouped in a monophyletic clade, including the MusGHV-1 virus, which likely belongs to the genus *Percavirus* [48]. A phylogenic analysis using the DNApol gene sequence showed that the novel and the common variant share a common ancestor (Figure 2). The Cornish variant (NC_038266) was derived from the common variant more recently, making the more recently discovered variant the actual ancestral variant.

### 3.3. Novel MusGHV-1 Variant Infections Were Clustered within Just Three Social Groups

MusGHV-1 variant frequencies did not differ significantly between males and females (Fisher’s exact test, *p* = 0.19) or between age groups (*p* = 0.19) but varied somewhat between social groups (*p* = 0.052) (Appendix A). The novel variant (and coinfection with both variants) was confined to badgers from three neighbouring social groups, while the common strain was evenly distributed across the Woods (Figure 3). To investigate further, we screened 14 DNA samples from blood collected from badgers resident in these three social groups during 2009 and 2010. Two individuals were infected with the novel variant, one was infected with both variants and the rest were infected with the common variant. This retrospective scrutiny indicated that the novel variant originated in the CHO social group before spreading to the CH and RC groups (Appendix A).

### 3.4. The MusGHV-1 Novel Variant Was Associated with Higher Viral Load in the Blood Stream

The Ct values from real-time PCR on 40 blood samples collected from 24 individuals in spring, summer and autumn 2018 (Table 2) showed that the MusGHV-1 relative viral load was highest (i.e., low Ct value) among badger cubs (age class) then decreased thereafter in yearlings (age class 1) (Figure 4). Furthermore, juveniles had higher relative viral loads than adults (*p* < 0.001), and samples taken in summer had higher loads than those in spring or autumn (*p* = 0.006) (Table 4). The mixed-effect linear regression analysis results suggest that badgers infected with the novel variant generally had a higher MusGHV-1 viral load than those infected with the common variant (*p* = 0.004).

## 4. Discussion

Understanding epidemiological traits can aid the interpretation of viral load data across wildlife species. Despite this study’s relatively small sample size, epidemiological variations were apparent in viral loads. Higher relative viral loads of MusGHV-1 in juveniles corroborated a previous study using animals from the same population [18], where primary herpes infection with a higher viral load typically occurs early in life [49]. Higher viral loads in summer were correlated with a higher prevalence of genital tract viral DNA reactivation [20]. Most intriguingly, we discovered that the relative blood viral load was also dependent on the MusGHV-1 variant. This could indicate that the two MusGHV-1 variants found in Wytham have not only genotypic but also phenotypic differences. A higher viral load could cause a higher pathogen burden and additional stress on the host immune system, which can lead to higher susceptibility to other pathogens [18]. Accordingly, we previously found that badgers infected with the novel variant, which had higher relative viral loads in this study, had a higher risk of infection with *Clostridium perfringens*, a zoonotic bacterial pathogen that can cause severe enteritis [50].

The genetic diversity of herpesviruses strains found within a single host species is typically very low, even between host populations separated by geographical barriers. For example, the gammaherpesvirus (LcaGHV-1) variants found in two geographically distinct Canadian lynx (*Lynx canadensis*) populations (from Maine, USA, and Newfoundland, Canada) are identical apart from just two synonymous substitutions across the 3.4 kb nucleotide sequences of its partial gB protein and DNApol gene [51]. Considering the generally low mutation rate of herpesviruses and their strong host cospeciation traits [50], the coexistence of two closely related gammaherpesvirus variants in the same population is unusual. A molecular analysis of the DNApol gene of twelve EHV-2 strains circulating in horse herds in Iceland that had been isolated for more than 1000 years, still showed 99% to 100% similarity across a region of 483 base pairs to seven EHV-2 strains from other European countries [52]. Furthermore, country-wide screening for MusGHV-1 in Irish badgers in 2019 showed that 691 base pair sequences from 23 individuals were 100% identical to the reference MusGHV-1 DNApol sequence [21]; all of which demonstrates the stability of species-specific herpesvirus DNApol genes.

Establishing whether the MusGHV-1 variant in the Wytham Woods population originated from introduction or evolution is challenging without a genome-wide analysis of mutations and sequencing results from other badger populations across the UK and nearby countries. Potentially, MusGHV-1 may have coevolved with badgers from a different population (likely subject to greater isolation by distance than the Cornish and Irish populations, e.g., continental Europe [53]) and then have been introduced into the Wytham population more recently, but at least before 2009 [54]. However, there are no historical records of badgers either colonising or being introduced into Wytham from other subpopulations, and badger genetic diversity in this population is low [55]; although any such introduced badger could, plausibly, have caused contagion with the novel variant and died without leaving any descendants. Ultimately, it seems unlikely that the novel variant originated from an ex situ source.

A phylogenetic analysis revealed that the node from/at the Cornwall variant branched from the ancestral MusGHV-1 root occurred earlier than the node at which either Wytham population branched which separated and evolved independently from an earlier time point. Although the common variant is dominant in the Wytham and the Irish populations, this does not infer that the novel variant emerged (branched) later than the common variant. It is also possible that the common variant became dominant by selectively outcompeting the novel variant, even if that novel strain branched earlier. For instance, if the higher virulence of the novel variant (rising from replicative advantage) caused higher host mortality prior to some immune adaptation, or triggered a more robust host immune response to eliminate the pathogen [56], then a less virulent variant (common) would spread among surviving badgers and become dominant. Some explanation based on balancing or fluctuating selection between specific variant advantages [57,58], or among host immune alleles [59], could account for occasional coinfection and why infections of the novel variants were limited to only a few social groups due to incomplete over-dominance.

The hypothesis that virulence may be a trade-off against transmission is debated [56,60]. Although a higher viral load can indicate more efficient within-host replication or higher virulence [60], various vertebrate and invertebrate models show that selection for an intermediate replication level can maximise pathogen fitness [61,62]. This may prevent the novel MusGHV-1 variant found in this study from being completely wiped out from the badger population. Although the high replication efficiency of the novel variant may result in it being outcompeted by the common variant due to host immune activities, this advantage may aid the novel variant to transmit and seed in the population through virus-shedding individuals. The lifelong infection trait of the herpesvirus may allow the novel variant to persist in a latency stage in the host at subclinical levels, possibly beyond the detection limit of PCR.

An alternative but not mutually exclusive explanation is that differences in host–pathogen resistance determined by immune-related genes, such as MHC genes, may be a causal factor for new strain emergence and the coexistence of multiple strains within a population. In badgers, Sin et al. (2014) found that individuals with specific MHC variants had different blood viral loads of MusGHV-1. Similarly, in horses, the EHV-1-induced downregulation of cell surface molecules expressed by MHC I is allele-dependent [63], and the extent to which MHC I is downregulated is EHV-1 strain-dependent [64]. In humans, herpesvirus infection has been linked to dementia [65], as some human leucocyte antigen (HLA; equivalent to MHC) alleles that protect against dementia have a high binding ability to a variety of human herpesvirus strains, suggesting that this link may be dependent on host variant [66]. Moreover, the subtypes of human herpesvirus 8 (HHV-8, or Kaposi’s sarcoma-associated herpesvirus, KSHV) are clustered by ethnic group, and infection with different subtypes affects the risk of Kaposi’s sarcoma occurring [67]. However, Wytham’s badgers engage extensively in extra-group mating. Therefore, if a specific MHC variant is more susceptible to the novel variant reactivation, it should spread quickly across social groups; however, this is contrary to our observations where the novel variant expanded from one to three social groups over nine badger cohorts (from 2009 to 2018). This would require a recessive MHC gene circulating within these social groups only expressed through inbreeding [42]. Currently, we lack the data to corroborate or refute this hypothesis.

## 5. Conclusions

In this study, we report a newly discovered variant of MusGHV-1 in badgers, indicating that a highly host-specialised pathogen maintains a certain degree of genetic diversity at the host-population level. This could illustrate an ongoing coevolution between badgers and MusGHV-1. Furthermore, we also found that variant variation in MusGHV-1 results in different epidemiological and spatial patterns of infection. By better understanding viral genetics and pathotypes and how these interact with coevolving host immunity, we can accurately inform conservation management and initiatives, such as One Health, tasked with disease surveillance and prevention in wildlife.

## Figures and Tables

**Figure 1 pathogens-11-01154-f001:**
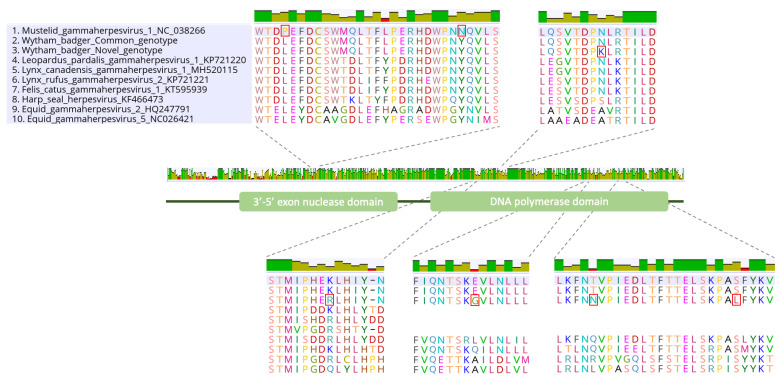
DNA polymerase gene alignment containing three MusGHV-1 variants and seven other gammaherpesviruses belonging to the genus *Percavirus*. Red rectangles show amino acid substitutions between the three MusGHV-1 variants (numbers 1 to 3 on the list).

**Figure 2 pathogens-11-01154-f002:**
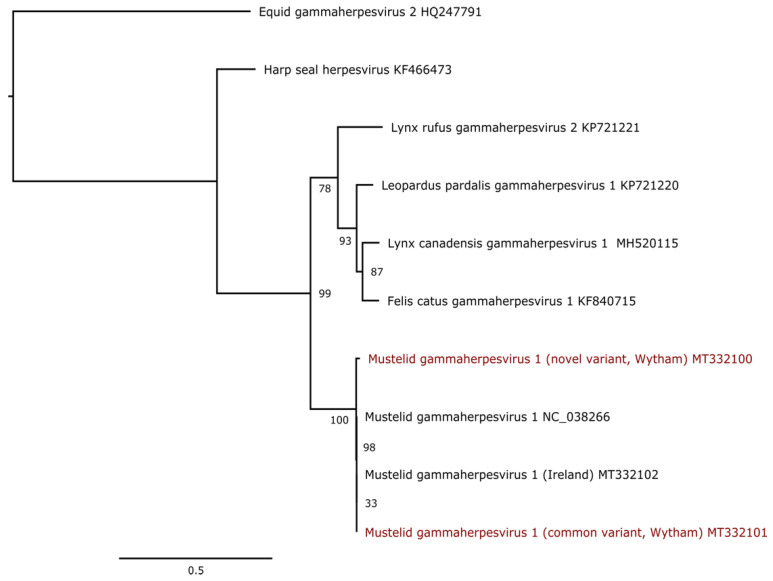
Maximum likelihood phylogenetic tree based on a portion of the polymerase gene of 10 gammaherpesvirus sequences. The alignment of sequences trimmed at 693 nucleotides was produced using MAFFT v7.450 using the G-INS-i algorithm. The tree is mid-point rooted. Bootstrap values (100 replicates) in excess of 30% are indicated at each node. The scale bar corresponds to nucleotide substitutions per site. The MusGHV-1 variants discussed in this paper (i.e., common and novel variants) are coloured in red.

**Figure 3 pathogens-11-01154-f003:**
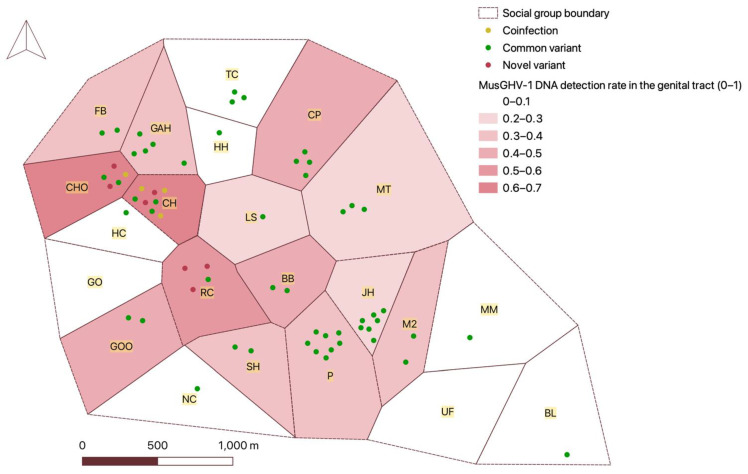
Map of genital tract MusGHV-1 occurrence and MusGHV-1 variants distribution in badger social groups in Wytham Woods in 2018. The name of each social group is highlighted in yellow. The areas with grey colour indicate sample sizes of less than 3.

**Figure 4 pathogens-11-01154-f004:**
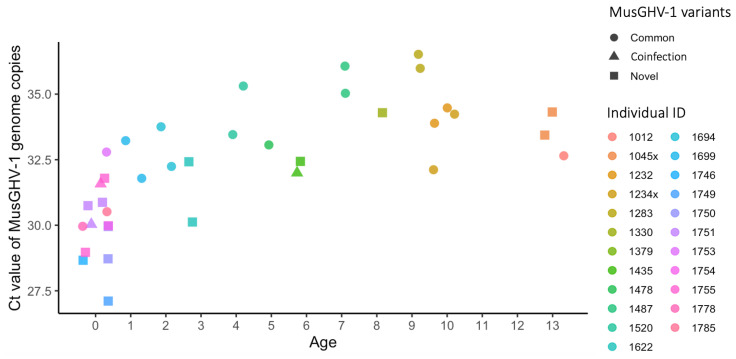
Ct values representing relative blood MusGHV-1 viral load of 39 blood samples collected from 23 individuals in different seasons (see Appendix A for details). The relative viral load was higher in juveniles (less than 2 years old) than in any other age class (linear mixed-effect model, *t*-test, *p* < 0.001, Table 4).

**Table 1 pathogens-11-01154-t001:** Primers used for MusGHV-1 DNA polymerase gene PCR amplification.

	Start Position	End Position	Primer Sequence (5′-3′)	Product Length
pol5-F	3	22	GGCAGGGAATTTTTATAACC	699
pol5-R	681	701	CCACCCAAAAGTAGAAAATCC	
pol6-F	345	364	CCCCTCTGGAACTGTGCTAA	637
pol6-R	962	981	AGGGTCACATGTCCCCAAAT	
pol7-F	863	846	ATGTCTGGGGGAAAATGG	486
pol7-R	1329	1348	GACCTCCTATGCACTGCTTG	
pol10-F	1185	1204	TGAAGTTCACACACCCCAGA	361
pol10-R	1524	1545	TCCATCGGTCAGCACTCTC	
pol3-F	1328	1347	CCAAGCAGTGCATAGGAGGT	771
pol3-R	2072	2098	TGGACTTCTCCAACATGCGTCGCCCTT	
pol11-F	1684	1703	CCGATCTTGGTGGTTGATTT	627
pol11-R	2291	2310	CTTAATTGGCTCCTCGAACA	
pol12-F	2015	2034	CAGGTGTGTCCTCGGGTATT	561
pol12-R	2556	2575	TCACTTTGAAAAGTGGAAGTGG	
pol9-F	2393	2412	TGATGAAGGGAGTGGATCTC	597
pol9-R	2936	2989	TCACAGCTTTGTCTGCACTG	

**Table 2 pathogens-11-01154-t002:** Blood samples used for relative viral load analysis.

Tattoo Number	Sex	Age	Spring	Summer	Autumn
1012	Male	13		v	
1045x	Female	13	v		v
1232	Female	10	v		v
1234x	Female	10		v	v
1283	Male	9	v	v	
1330	Male	8		v	
1379	Male	>8 *	v		v
1435	Female	6		v	v
1478	Female	5		v	
1487	Male	7	v		v
1520	Female	4	v	v	
1622	Male	3	v	v	
1694	Female	2		v	v
1699	Male	1		v	v
1735	Male	0	v		
1746	Male	0		v	
1749	Female	0	v		
1750	Female	0		v	v
1751	Female	0	v	v	v
1753	Female	0	v		
1754	Male	0	v		
1755	Male	0	v	v	v
1778	Male	0		v	
1785	Female	0			v

* Age not determined as this individual was first captured as an adult; according to its tooth wear [43], it was estimated as older than 8 years of age.

**Table 3 pathogens-11-01154-t003:** Amino acid substitutions (nonsynonymous differences) between the two variants in the MusGHV-1 DNA polymerase gene.

Position	Reference (AF275657 ^a^)	Common Variant	Novel Variant	Conserved Protein Domain Family ^b^	In Conserved Region ^c^	In Conserved Region within Host Order ^d^
253	P	L	L	3′-5′ exonuclease Exo I	Yes	Yes
274	N	Y	Y	3′-5′ exonuclease Exo I	Yes	Yes
591	K	K	R	Polymerase	No	No
649	N	N	K	Polymerase	No	Yes
819	E	E	G	Polymerase	No	No
872	T	T	N	Polymerase	No	No
889	S	S	L	Polymerase	No	Yes

a: GenBank accession number. b: Data from the NCBI Conserved Domain Database (CDD) [46]. c: Determined by whether the substitution is located in previously identified conserved motif of viral DNA polymerase genes [47]. d: Determined by whether the substitution is located in a position with a higher conservation degree within herpesvirus species confirmed in the order Carnivora (Figure 1).

**Table 4 pathogens-11-01154-t004:** Results of linear mixed-effect model fit by REML (restricted maximum likelihood). *t*-tests using Satterthwaite’s method (formula= Ct ~ Variant + Age Group + Season + (1 | Tattoo)). Number of observations: 39, groups (by tattoo ID: 23).

Variable		Estimate	Std. Error	df	*t* Value	*p* Value
(Intercept)		34.855	0.464	28.794	75.072	
Genotype						
	Common	(Reference)				
	Novel	−1.815	0.547	19.447	−3.316	0.004
	Coinfection	−0.346	0.834	32.885	−0.415	0.681
Age Group						
	Adult	(Reference)				
	Juvenile	−3.030	0.543	18.615	−5.583	<0.001
Season						
	Spring	(Reference)				
	Summer	−1.379	0.453	23.179	−3.045	0.006
	Autumn	0.012	0.416	17.381	0.028	0.978

## Data Availability

Not applicable.

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
