# Peer review of "Infection with a Recently Discovered Gammaherpesvirus Variant in European Badgers, Meles meles, is Associated with Higher Relative Viral Loads in Blood"

_pathogens, 2022, doi:10.3390/pathogens11101154_

Round 1

Reviewer 1 Report

Dear Colleagues,

The paper is well written and very interesting. 

Just a few issues (only issue 1 is relevant, while all the others are minor):

1 - I cannot find any figure representing the phylogenetic tree of the viral variants, neither in the main text, nor in the supplementary material. I think it should be added

2 - Row 224: Probably the Figure should be renamed as S2 in the supplementary material

3 - Row 272-275: It is unclear wether a cause-effect relation is supposed between the novel variant and enteritis due to C. perfrigens. Sorry to have no time to read the paper you cited on this topic. However, I would consider the alternative (but not mutually exclusive) of stress/immunosuppression as the common origin of both. Could you please clarify your hypothesis?

4 - It is not clear for me wheter you consider the hypothesis of an ecological role of this virus, and in general of pathogens strictly co-evolving with their host species, in maintaining a certain degree of genetic variability within the host population.

I hope you'll find my comments useful.

Sincerely

Author Response

Thank you so much for your time reviewing our manuscript as well as for your thoughtful comments. Please see our reply in red to each point you listed in the report.

1 - I cannot find any figure representing the phylogenetic tree of the viral variants, neither in the main text nor the supplementary material. I think it should be added

We apologize for omitting the phylogenetic tree in the supplementary materials. We attached a wrong SM version with the initial submission. We have added the phylogenic tree to the main text as Figure 2 and revised the materials and methods. (L147-150)

2 - Row 224: Probably the Figure should be renamed as S2 in the supplementary material

We have changed this to Figure 2 as we decided to put this in the main text. (L233)

3 - Row 272-275: It is unclear wether a cause-effect relation is supposed between the novel variant and enteritis due to C. perfrigens. Sorry to have no time to read the paper you cited on this topic. However, I would consider the alternative (but not mutually exclusive) of stress/immunosuppression as the common origin of both. Could you please clarify your hypothesis?

Thank you for pointing this out. We have added another sentence to explain the hypothesis to improve clarity according to your suggestion. (L295-298)

4 - It is not clear for me wheter you consider the hypothesis of an ecological role of this virus, and in general of pathogens strictly co-evolving with their host species, in maintaining a certain degree of genetic variability within the host population.

Thanks for the comment. We have revised the conclusion to clarify the link between our results and these hypotheses. (L381-384)

Reviewer 2 Report

This is a well written and presented paper that can be published in its present form

Author Response

Thank you so much for your generous comment and time reviewing our manuscript.